# Novel Approaches to Improve Meat Products’ Healthy Characteristics: A Review on Lipids, Salts, and Nitrites

**DOI:** 10.3390/foods12152962

**Published:** 2023-08-05

**Authors:** Sandra S. Q. Rodrigues, Lia Vasconcelos, Ana Leite, Iasmin Ferreira, Etelvina Pereira, Alfredo Teixeira

**Affiliations:** 1Centro de Investigação de Montanha (CIMO), Instituto Politécnico de Bragança, Campus de Santa Apolónia, 5300-253 Bragança, Portugal; lia.vasconcelos@ipb.pt (L.V.); anaisabel.leite@ipb.pt (A.L.); iasmin@ipb.pt (I.F.); etelvina@ipb.pt (E.P.); teixeira@ipb.pt (A.T.); 2Laboratório Para a Sustentabilidade e Tecnologia em Regiões de Montanha (SusTEC), Instituto Politécnico de Bragança, Campus de Santa Apolónia, 5300-253 Bragança, Portugal

**Keywords:** meat products, lipids, salts, nitrites, new technology, new processes, reduction/replacement, health

## Abstract

Meat products are a staple of many diets around the world, but they have been subject to criticism due to their potential negative impact on human health. In recent years, there has been a growing interest in developing novel approaches to improve the healthy characteristics of meat products, with a particular focus on reducing the levels of harmful salts, lipids, and nitrites. This review aims to provide an overview of the latest research on the various methods being developed to address these issues, including the use of alternative salts, lipid-reducing techniques, and natural nitrite alternatives. By exploring these innovative approaches, we can gain a better understanding of the potential for improving the nutritional value of meat products, while also meeting the demands of consumers who are increasingly concerned about their health and well-being.

## 1. Introduction

Meat, when consumed as sustenance, refers to the flesh of animals. Ingestion of meat by early humans began more than 2 million years ago and it is believed that meat contributed for cerebral evolution due to its high content in energy and nutrients. Meat was consumed as the animals were being hunted and slaughtered, and there was no capacity to store it for long periods. To preserve meat for longer periods, humans started processing meat products. Nowadays, processing meat can also provide differentiated products resulting in the valorization of some less valued types of meat. Processed meat, in general, denotes meat that has undergone preservation methods such as smoking, curing, salting, drying, or other techniques aimed at enhancing flavor, quality, or extending its shelf life. Additionally, processed meat may contain poultry, offal, or ground meat byproducts. Illustrative examples of processed meat encompass ham, bacon, pastrami, salami, and sausages [1].

The ingestion of meat and meat-based products furnishes the human body with a valuable array of vital nourishing elements, such as proteins, iron, and various vitamins. These substances are crucial for the proper functioning and development of the human body, serving as essential components within a varied diet. Meat guarantees the sufficient provision of necessary micronutrients and amino acids that play integral roles in the regulation of energy metabolism, promoting human health and growth. Nevertheless, despite its status as a fundamental and exceptional source of top-quality proteins, iron, zinc, and vitamin B12, the consumption of red meat (meat from animals with higher content of red muscle fibers), and particularly processed meat, has been linked to an elevated susceptibility to chronic ailments and pathological conditions [2], as cardiovascular disease due to its saturated fat and cholesterol content, or cancer [3,4,5], as colorectal esophageal, gastric, bladder, and other cancers, due to its high protein content or type of processing (e.g., heat, smoke) [6].

Additionally, during the processing of meat, additional meat and animal fat may be added together with a wide range of non-meat substances and additives, leading to more complex products and different impacts on health. In October 2015, the World Health Organization (WHO) International Agency for Research on Cancer (IARC) announced that consumption of processed meat is “carcinogenic to humans” (Group I) and that consumption of red meat is “probably carcinogenic to humans” (Group 2A). A report from the World Cancer Research Fund and the American Institute of Cancer Research (WCRF/AICR) based on an updated systematic review of previous research concluded that there is strong evidence, mostly from Asia, that consuming foods preserved by salting (including salt-preserved vegetables, fish, and salt-preserved foods in general) is a cause of stomach cancer, and that high intake of red meat probably and processed meat convincingly increases the risk of colorectal cancer [7]. Excessive salt intake represents a risk factor for the onset and/or progression of several chronic diseases, such as cardiovascular diseases and various types of cancer [8,9]. Meat and meat products for some consumers have been considered as potentially unhealthy foods due to their nutritional characteristics associated with high salt and fat content [10,11,12]. Additionally, it was found that higher nitrite and nitrate intake from processed meats contributes to the increase in diastolic blood pressure [13].

In recent years, there has been an increasing interest in enhancing the health attributes of meat products to meet the growing demand for healthier food choices. To address this demand, researchers have explored novel approaches aimed at improving the nutritional profile of meat, with a particular focus on lipids, salts, and nitrites–nitrates.

One area of investigation involves the modification of the lipid composition of meat products. Saturated fats, commonly found in meat, have been associated with increased risk of cardiovascular diseases. As a result, researchers have focused on incorporating healthier lipid sources, such as unsaturated fats and omega-3 fatty acids, into meat formulations. These modifications not only can improve the nutritional profile but also enhance sensory attributes and oxidative stability [14,15,16,17,18,19,20,21,22,23,24,25].

Furthermore, attention has been given to the reduction in sodium chloride (salt) content in meat products. Excessive consumption of salt has been linked to various health issues, including hypertension. Therefore, researchers have been exploring alternative salt replacers, such as potassium chloride, calcium chloride, and magnesium chloride, to maintain the desired taste and functionality of meat products while reducing sodium levels [26,27,28].

Additionally, the use of nitrites in meat products has received considerable attention. Nitrites play a crucial role in preserving meat color, flavor, and inhibiting bacterial growth. However, concerns have been raised regarding their potential health risks, including the formation of carcinogenic compounds. Therefore, alternative strategies, such as the use of natural antioxidants and antimicrobials, have been explored to replace or reduce the reliance on nitrites in meat processing [29,30,31,32,33].

Even though meat products follow health recommendations, they also need to fulfill consumer demands for organoleptic characteristics. Organoleptic characteristics refer to the sensory attributes of food, including its appearance, taste, aroma, texture, and overall palatability. These qualities play a crucial role in consumers’ perceptions and preferences concerning meat products. Consumers utilize a combination of sensory properties and other extrinsic factors to predict and assess beef quality [34]. The sensory characteristics of beef products play a central role in determining consumer preferences and willingness to pay [35]. Results from Pellatiero [36] show that the type of meat used, related to its fat content, can modify consumers’ sensory perception of processed meat products. According to existing research, the process for creating healthier processed meat products involves a significant reformulation process that can lead to alterations in sensory characteristics [37]. Therefore, leveraging sensory science can be a crucial approach to foster the development of healthier processed meat products that have higher prospects of market success [38].

This review examines the scientific literature and recent studies on these novel approaches, discussing their efficacy, challenges, and potential impact on the overall healthiness of meat products. By understanding and implementing these advancements, the meat industry can contribute to providing consumers with healthier and more nutritious choices without compromising taste or quality. Focus is given to lipids, salts, and nitrites/nitrates.

## 2. Approaches towards Healthier Meat Products

### 2.1. Lipids in Healthier Meat Products

Meat products play a significant nutritious role in our diet, although some ingredients have been linked to unfavorable health effects. The improvement of lipid content has drawn considerable attention among the new trends in the design of healthy meat products, which primarily concentrate on enhancing their composition in quantitative and qualitative ways. Lipids represent an important energy source and assist in nutrient absorption, such as fat-soluble vitamins; apart from that, its consumption should be balanced [39,40].

A high fat intake represents a risk factor for diseases such as obesity, heart conditions, high blood pressure, diabetes, and others; in contrast, there is evidence that consumption of polyunsaturated (PUFAs) and monounsaturated (MUFAs) fats have a beneficial role in health [41,42]. Dietary recommendations for fat intake establish no more than 10% of saturated (SFAs) fats, 25% of unsaturated (UFAs) fats and less than 1% of trans fatty acids [41]. To meet these recommendations and improve the lipid content of meat products, technological strategies generally replace animal fat with different lipids to achieve healthier characteristics without major changes in the product’s properties, not only for fat reduction, but also focusing on the optimization of lipidic quality and inclusion of functional ingredients [42,43,44].

Fortifying processed meat products with compounds that may mitigate or neutralize such adverse health effects is one strategy that has been used to prevent detrimental effects linked to the consumption of processed meat. However, physical and thermal properties should be similar to animal fat in order to completely replace it in food products, since the fat contributes to several technological traits in meat products, with variable effects depending on product specifications [45]. The strategies include the replacement of animal fat by vegetable oils, the use of flours and fibers, hydrocolloids, mushrooms, and some animal proteins such as whey and collagen. Although the replacement of animal fat by liquid oils generates primarily textural issues, affecting mouth-feel, juiciness, and fat binding properties, demanding an additional step and components to mimic the fat structure and minimize the impacts of the replacement [43].

Given the difficulty and significance in maintaining meat products quality while replacing fat, some methods to lower the amount of animal fat in meat products or increase the quality of lower-fat meat products consist in substituting animal fat with UFA-enriched lipid sources; the other focus is adapting low-fat formulations using physical methods to minimize the impact caused by fat reduction [46].

On the other hand, there are physical methods that comprise high pressure processing (HPP). During the non-thermal process, a partial denaturation of protein molecules caused by the water molecules being pressed into the muscle fibers reduces the cross-linking between non-polar groups, promoting the interaction between proteins and water molecules [46,47]. This technology acts mainly to improve textural problems caused by fat reduction, maintaining tenderness, juiciness, and chewiness as similar as possible to the traditional products. Another physical method is ultrasonic technology, which uses cavitation phenomena to alter intermolecular interactions for the formation of a gel, being used for improving color parameters, water-holding capacity, and texture [46]. In Table 1, it is possible to observe some examples of the studies conducted utilizing the approaches mentioned and how the technologies affected the products.

Overall, a reduction in total fat, and an increase in mono- and polyunsaturated fatty acids, mainly oleic and linoleic acids, was observed. Implications on physicochemical, rheological, and sensory characteristics varied among the studied products. In some cases, the alteration in the type of fat used in the production of the meat product increased, others decreased, and others did not affect the meat product attributes. This means that while trying to improve the product’s lipid profile, we can aggravate the quality of other attributes. As an example, changes in fat addition caused an increase in lipid oxidation, which means that the products become more perishable [17,48,49]. Hardness increased in goat burgers [50] and beef burgers [51], which can be a less-appreciated characteristic for consumers.

**Table 1 foods-12-02962-t001:** Technological approaches and their results to reduce or replace fat in different meat products.

Product	Technology	Results	Reference
Beef burgers	Gelled emulsion with soybean oil, Maca flour, and Chincho essential oil	Reduced fat, proteins, and SFA contents, increased PUFA and PUFA/SFA ratio. Hardness decreased and lipid oxidation increased.	[48]
Bologna sausage	Gelled emulsion with soybean	Reduced total fat and SFA content. Effects in physicochemical, rheological, and microstructural properties were greater with reformulation.	[52]
Alheira	Gelled emulsion based on hemp oil and buckwheat	Increased linoleic and linolenic acids, reduced SFA contents. Physicochemical characteristics were maintained. There was an increase in lipid oxidation related to replacement levels.	[49]
Goat burger	Oleogel with olive oil	Reduced total fat, increased MUFA and PUFA contents. Related to color, redness, and yellowness were higher for burgers with oleogel. Sensory analysis indicated an increase in hardness and chewiness of goat burgers.	[50]
Beef burger	Oleogel with pork skin and olive oil	Reduced fat and energy contents, there was an increase in protein content and the FA profile was improved by oleogel use. Hardness and chewiness were higher than the control and the lipidic oxidation remained stable for 7 days.	[51]
Lamb sausage	Oleogel with lecithin or sorbitol monostearate and canola oil	Increased UFA/SFA ratio and reduction in lipidic oxidation. Decrease in cooking loss, hardness, springiness, chewiness, and resilience. Sensory scores did not show difference up to 50% fat replacement.	[53]
Goat burger	Hydrogel of sunflower and olive oils	Fat content was reduced, FA profile improved, with SFA content reduction, and increased MUFA and PUFA, specifically oleic and linoleic acid content.	[54]
Pork sausage	Hybrid hydrogel of inulin and microcrystalline cellulose	Reduced total fat content and peroxide value, although it increased lipidic oxidation. Hardness, chewiness, and water-holding capacity were improved.	[55]
Deer pate	Microencapsulation of tigernut, chia, and linseed oils	Reduced fat and cholesterol contents, decrease in SFA and increase in PUFA and MUFA contents. Texture was considered softer than control; for color, there were higher values for redness and yellowness. Lipidic oxidation was higher in pates with high n-3 PUFA levels.	[17]
Pork sausage	High-pressure treating time in reduced fat and reduced salt	Reduction in fat and salt, and moisture increase. Higher water-holding capacity and tenderness, reduction in hardness.	[56]
Pork emulsion sausages	Ultrasound and aminoacids (L-lysine and L-arginine)	Combination resulted in better outcomes, with a reduction in cooking loss, expressible fluids, and fat content.	[57]
Chicken nuggets	Control chicken nugget with chicken skin substitution by chia (*Salvia hispanica* L.) flour 5, 10, 15, and 20%.	5% chia flour similar to control for aroma and flavor attributes; chia flour >10% impaired acceptance for all evaluated attributes.	[58]
Chicken nuggets	Fish oil (FOL) and encapsulated fish oil by tragacanth (TRG) and carrageenan (CGN)	Fish oil encapsulation exhibited a safeguarding effect against lipid and protein oxidation. Enhancement of nuggets’ oxidative shelf life and sensory attributes. Tragacanth for encapsulating the fish oil was more efficient in maintaining the sensory characteristics.	[59]
Chicken nuggets	Okara flour (OF) and rice bran (RB) as fat substitutes	Increased dietary fibers, ashes, and PUFAs. Reduced lipids content, saturated fatty acids, and cholesterol. Better sensory acceptance and higher purchase intention when mixing OF and RB.	[60]

### 2.2. Salts on Healthier Meat Products

For millennia, salt has been a key food preservative. Its use can be traced back thousands of years, highlighting its longstanding role in ensuring food safety and extending the shelf life of various food products. The antibacterial properties of salt, in particular their ability to inhibit bacterial growth, have made it an indispensable tool in food preservation techniques across different cultures and civilizations. Even in modern times, salt continues to contribute a significant role in preserving the quality and safety of food. Its effectiveness as a preservative has stood the test of time, making it a staple ingredient in numerous culinary traditions worldwide. In addition to that, sodium plays a crucial role as a vital mineral in maintaining blood volume and pressure. It is recommended that individuals consume up to 5 g of sodium chloride (NaCl) per day [61], as around 90% of the sodium in our diets comes in this form [62]. Unfortunately, the current situation reveals an excessive intake of sodium, primarily through table salt (NaCl), which can elevate the risk of cardiovascular diseases [63].

Sodium has several functions as a food ingredient (curing meat, cooking, thickening, retaining moisture, enhancing flavor, and acting as a preservative). Sodium is present in common food additives such as monosodium glutamate (MSG), sodium bicarbonate (baking soda), sodium nitrite, and sodium benzoate. Although some foods that are not considered salty may be high in sodium, taste alone is not a reliable indicator of sodium levels in foods. In addition, certain foods such as breads (eaten daily) can contribute significantly to total sodium intake, even if an individual serving is not particularly high in sodium. Consumers have the ability to swiftly verify the sodium content of the products they purchase, enabling them to make informed choices by selecting products with reduced sodium levels. Product labeling may feature various claims related to sodium content, such as “no salt/sodium,” “very low sodium,” “low sodium,” “reduced sodium,” “low sodium or slightly salty,” and “no salt added” or “no salt” [64].

One practical approach to reducing sodium is the gradual decrease in its presence over time, giving consumers the possibility to adapt to lower levels of NaCl in their food [65,66]. However, implementing this strategy requires consensus and cooperation between the food industry and health authorities. The food industry has prioritized the reduction in sodium levels in processed foods. According to the WHO, meat products play a significant role in daily sodium intake, accounting for approximately 16–25% of the total. They are considered the second-largest contributor of sodium in the diet, following bakery products. However, lowering sodium content in meat products presents significant challenges due to the vital technological, sensory, and microbiological stability properties provided by sodium chloride (NaCl).

NaCl, for instance, acts as a suppressant in microbial growth, [67,68], helping to maintain product safety. It also plays a role in inhibiting the activity of proteolytic enzymes, which can affect the physicochemical and sensory characteristics of meat products [69]. Moreover, NaCl influences lipid oxidation and lipolysis reactions [67], which impact the flavor and quality of the final product. These properties make it difficult to reduce sodium levels without compromising the desired attributes of meat products [70]. Efforts to reduce sodium in meat products require careful consideration of alternative strategies and ingredients that can provide similar technological functionalities while maintaining sensory acceptance and microbiological stability. Balancing these factors is crucial to ensure that reduced-sodium meat products meet consumer expectations without compromising safety or quality [22].

The production of salted meat products involves several salting steps, which are of extreme importance for obtaining the desired characteristics of the final product. The salting methods employed vary depending on the specific product and its intended purpose, directly influencing its properties. The chosen salting method determines the mechanism of mass transfer and whether there will be an increase or decrease in weight during the process. These factors play a significant role in shaping the final texture, flavor, and preservation of the salted meat. The primary objective of salted meat processing is to lower the water activity to levels that ensure microbiological stability at ambient temperatures. By reducing the activity of water, the growth and proliferation of microorganisms are inhibited, thereby extending the shelf life of the product. Overall, careful implementation of salting techniques is essential in the production of salted meat products, as it directly impacts their quality, preservation, and safety [66].

In many meat products, a significant amount of NaCl is added to improve texture, facilitate emulsion formation, increase production yield, inhibit microbial growth, and provide distinctive sensory characteristics [10,66]. Hence, any modification in meat products concerning reducing salt content must be accompanied by a comprehensive evaluation of their sensory acceptance, physicochemical properties, and stability. Many studies are focused on ways to reduce or substitute sodium that can provide sensorial and technological functions like NaCl: for example, replacing NaCl with other chlorine salts (such as KCl, CaCl_2_, and MgCl_2_) [71,72]; replacing for non-chloride salts as lactates and phosphates (such as K-lactate and Ca-lactate) [73]; using flavor enhancers such as taurine, lysine, or monosodium glutamate [74]; incorporating natural products with a salty taste such as yeast extract, hydrolyzed vegetable protein, and seaweeds [66]; or applying novel techniques such as high pressure and ultrasonic methods [75,76,77,78]. The use of ultrasound is based on an understanding of mass transfer processes and how these can modify cell membranes, aiding the various stages in the curing process [79]. Ultrasound is one of the main technologies used in processed meat industry, allowing an increase in shelf-life, thus prolonging the flavor, juiciness, and tenderness of the products for the final consumer. For example, McDonnel et al. [80] studied the possible industrial application of ultrasound-cured ham and used pork meat samples that were treated with different ultrasonic intensities; 40, 56, or 72 W cm^2^ for 2, 4, or 6 h, respectively. In all the samples, the desired level of NaCl (2.25%) was reached within 2 h, while the control (employing no ultrasound) required 4 h. Sonication showed no negative effect on cooking loss, moisture, or texture profile. Sensory analysis revealed a positive correlation between the product flavor and a stronger ultrasound power application.

However, despite many studies having shown that partially replacing NaCl with KCl is the most used/common choice for producing low-sodium meat products [81,82], replacing of 50% NaCl with KCl would have negative sensory properties such as a bitter metallic taste and astringency [83,84]. Additionally, the application of basic amino acids including L-lysine (L-Lys), L-histidine (L-His), and L-arginine (L-Arg) has aroused considerable interest. These amino acids can act independently or synergistically because they contain an extra amino group where their positively charged side chains at pH 7 can interact with groups with opposite charges, forming ionic bonds or salt bridges [85]. Thus, they are used as substitutes in ham [86], dry-cured loin [82], cured and cooked loin [87], and sausages [22,88]. In another study [89], salted meat treated with 3% L-Lys + (50% NaCl, 25% KCl, and 25% CaCl2) had a higher overall acceptance compared with the sample treated with only 50% NaCl, 25% KCl, and 25% CaCl_2_ (i.e., without L-Lys).

Consequently, reducing or substituting NaCl directly with other salts poses a challenging technological task. Furthermore, the demand for clean label foods restricts the use of synthetic ingredients, and the high production costs associated with these ingredients complicate NaCl substitution [90].

Table 2 summarizes some approaches, tools, and implications in reducing or replacing salts in meat products to make them healthier.

### 2.3. Nitrites/Nitrates on Healthier Meat Products

The utilization of nitrite and nitrate in meat products has become a topic of heightened scrutiny in both meat science research and the processed meat industry, with a focus on promoting healthier options. However, their inclusion can still offer certain benefits. Nitrite and nitrate are often utilized in cured meat products due to their antimicrobial properties [99], which can help inhibit the growth of harmful bacteria such as *Clostridium botulinum* and *Listeria monocytogenes* [100]. These compounds also exhibit antioxidant activity, which aids in reducing the risk of meat product rancidity [101] and also promotes the development of the reddish-pink color and the flavor characteristics of cured meat products [102]. However, it is crucial to emphasize that the quantities of nitrite and nitrate added must be closely monitored to ensure consumer safety. Excessive consumption of these compounds carries a high risk of certain cancerous diseases [103], as sodium nitrite can serve as a precursor for the formation of carcinogenic compounds such as nitrosamines [104]. The main nitrosamines found in meat products are N-nitrosodimethylamine (NDMA), N-nitrosodiethylamine (NDEA), N-nitrosopiperidine (NPIP), N-nitrosopyrrolidine (NPYR), and N-nitrosomorpholine (NMOR) [103]. The application of nitrite in the meat industry has been a matter of concern for both industry professionals and consumers over the years [105]. As the risk of colorectal cancer associated with the consumption of red meat and processed meat has been acknowledged, it is essential to explore the underlying mechanisms responsible for this link [106]. The cited authors elucidated the evidence on the catalytic effect of heme iron on the endogenous formation of carcinogenic N-nitroso compounds and on the production of cytotoxic and genotoxic aldehydes through lipoperoxidation. Previous investigations on the relationship between processed meat and colorectal cancer have already highlighted that heme iron present in red meat can promote carcinogenesis by enhancing cell proliferation in the mucosa, potentially through lipoperoxidation and/or cytotoxicity of fecal water [107]. According to the cited authors, nitrosation has the potential to increase heme toxicity in cured products. Tackling and comprehending the mechanism of nitrosation is a challenging endeavor that offers the opportunity to lower cancer risks through process modifications rather than resorting to an outright ban on processed meat consumption. Undoubtedly, for today and in the near future, this is one of the most important research topics for production, industry, and consumers that concerns improving meat products’ health effects [108,109,110,111].

Considering these concerns, governmental bodies and health institutions have established threshold limits for nitrite consumption. Regulation (EC) No. 1333/2008 [112] of the European Parliament and of the Council of 16 December 2008 on food additives in its Annex II, part E, food category 08.3 “Meat products”, sets maximum levels for potassium nitrite (E 249) and sodium nitrite (E 250) that may be added during manufacture. These maximum levels have been set at 150 mg/kg for meat products in general and 100 mg/kg for sterilized meat products. For a few specific cured meat products traditionally manufactured in certain Member States, the maximum added level is set at 180 mg/kg. Consequently, it becomes imperative to test the nitrite levels in meat products to ensure food safety and safeguard consumers against potential health issues [113]. In a study conducted in fermented sausages processed with different rates of sodium nitrite and sodium nitrate, [108] concluded that the minimum rate of 80/80 ppm nitrite/nitrate was sufficient to provide protection against lipid oxidation in the digestive tract. A food composition database was developed to assess nitrate and nitrite intake from animal-based foods, aiming to investigate the associations between dietary nitrate and nitrite intake and health outcomes [111]. Alternative approaches and ingredients are being explored to reduce the reliance on nitrite and nitrate while still achieving the desired flavor, color, and safety aspects in healthier meat products. Ongoing research aims to identify innovative solutions that balance health considerations with the preservation and sensory attributes of cured meats.

In recent years, there has been a growing discussion about the utilization of natural sources of nitrite [114]. Additionally, strategies were addressed [115] for enhancing the nutritional quality of meat and meat products, highlighting the reduction in nitrite levels as a means for imparting healthier characteristics to these products. Strategies have been explored to achieve this, including the use of natural sources of nitrite, as well as alternative methods and ingredients to preserve the safety, flavor, and overall quality of meat products. Ongoing research and discussions among experts in the field continue to focus on finding innovative approaches to effectively reduce nitrite levels while maintaining the desired characteristics of meat products. Also, the substitution of nitrite with alternative natural ingredients in meat products has gained considerable interest in recent years, regarding the potential health risks associated with nitrite consumption while still ensuring the safety and quality of the final products. Various natural ingredients have been explored as potential replacements for nitrite, such as celery powder, beetroot powder, and sea salt, among others [31,101,116,117,118,119,120]. These ingredients contain naturally occurring nitrates, which can be converted into nitrites during the curing process. By utilizing these natural sources, manufacturers can maintain the desirable antimicrobial and color-preserving properties traditionally associated with nitrite. However, it is essential to note that the use of natural alternatives requires careful formulation and precise control to achieve consistent results. Even if the impact of using nitrate or nitrite as curing ingredients on public health and the sensory properties, particularly the flavor perceived by consumers, has been extensively documented in scientific literature, there is a lack of understanding regarding consumer perceptions of meat products in which nitrite or nitrate has been reduced or replaced by alternative ingredients [37]. The sensory aspects, including flavor and appearance, need to be evaluated to ensure that the final products meet consumer expectations. Researchers, industry professionals, and regulatory bodies are actively involved in studying and developing guidelines for the effective and safe utilization of natural ingredients as nitrite substitutes. Ongoing efforts aim to strike a balance between health considerations and maintaining the sensory characteristics that consumers associate with cured meat products. Table 3 presents the sensory implications of studies focusing on product reformulations aimed at enhancing the healthiness of meat products by alterations in the usage of nitrites and nitrates.

By transitioning to nitrate and nitrite substitutes, we can preserve the safety of meat products while minimizing the potential harm associated with these compounds. Nitrate and nitrite substitutes offer a promising solution by reducing or eliminating the formation of nitrosamines while maintaining the desired properties of processed meats, providing a healthier alternative that does not compromise the taste, appearance, or shelf life of meat products. Moreover, nitrate and nitrite substitutes can offer additional health benefits. By incorporating these substitutes, we can introduce healthier options to consumers, promoting a well-rounded and balanced diet. As the demand for healthier and more natural food options continues to rise, it is essential to keep consumers informed about the use of nitrate and nitrite substitutes in meat products; providing transparent labeling and accurate information empowers consumers to make informed choices and encourages the industry to prioritize their health. The availability of a growing number of “clean” label ingredients provides a new suite of approaches that are available for application by meat processors to help overcome some of the negative connotations associated with processed meat products [120,126,127].

According to a review on “Natural alternatives for processed meat: legislation, markets, consumers, opportunities and challenges” [128], consumers’ interest in food with less and/or free from synthetic additives has increased considerably in recent years. Consumers are willing to pay more for healthier meat products from companies that add this information on the packaging. This trend has induced the meat industry to innovate, promoting new processing methods and technologies to meet new consumer expectations and the trend toward “clean label” foods. The impact of this trend on the market is that companies are seeking ways to incorporate natural alternatives into their products to meet consumer demand. Still, the industry faces some challenges when trying to incorporate natural alternatives into their products, given their importance for sensory characteristics and food safety. A successful reformulation to remove artificial preservatives involves a multifunctional combination between food safety, product development, technology modifications, and ingredient research. It is also important to note that a global reformulation of products will lead to an increase in production costs and, consequently, in the final cost of the food since chemical ingredients and additives are less expensive. However, with a higher production of clean-label foods, new suppliers of ingredients and additives will emerge in the market, and consequently, their prices will be reduced.

Aromatic herbs and essential oils, yeast extracts, and seaweed, among others, are examples of natural ingredients that can be used as substitutes for synthetic additives. The use of aromatic herbs in food products is exempt from mandatory nutrition declarations for labeling purposes, and there are no regulated maximum admissible limits for the use of this type of ingredient according to Regulation (EU) No. 1169/2011 [129].

## 3. Final Remarks

While health considerations are important, consumers also value the sensory experience when consuming meat products. Food manufacturers and producers must strike a balance between meeting health recommendations, such as reducing saturated fats and sodium content, while maintaining the desired organoleptic characteristics. Moreover, consumer demands are evolving, and there is an increasing interest in meat alternatives or plant-based meat products. These alternatives strive to replicate the taste, texture, and appearance of traditional meat products, catering to individuals who follow vegetarian, vegan, or flexitarian diets. Meeting these demands requires innovation in product development and technology.

The availability and price of the replacer products together with the costs of new processing methodologies are also very important and must be considered, as they can prevent the industry from applying them regardless of their potential to provide healthier meat products, and still be pleasant for consumers. Naturally abundant ingredients are often used as replacements for synthetic additives that are used to preserve or change the characteristics of some meat products. Even though it is a natural ingredient, an example, from personal experience, the use of *Salicornia* powder as a salt substitute is prohibitive to produce healthier goat burgers. This can be related to the processes needed to obtain the specific extract or nutrient, or even the physical state needed for its proper use.

## 4. Conclusions

Meat products play a significant role in our diet, but their high fat content, especially saturated fats, can pose health risks. To address this, there is a growing focus on improving the lipid content of meat products by replacing animal fat with healthier alternatives. Additionally, fortifying processed meats with compounds to mitigate adverse health effects is a strategy being employed. These efforts aim to create healthier meat products without compromising their desirable properties.

Alternative strategies to reduce salt in meat products have shown promising results. Partially replacing sodium chloride (NaCl) with potassium chloride (KCl) is a common approach, although it may lead to negative sensory properties. The use of basic amino acids such as L-lysine, L-histidine, and L-arginine has also been explored and found to enhance sensory acceptance. Additionally, the application of novel techniques like high pressure and ultrasonic methods has shown potential in enhancing salt distribution and perception while reducing overall NaCl content. These alternative strategies aim to maintain product quality and safety while reducing sodium levels.

Replacing nitrite and nitrate in meat products with natural alternatives like celery powder and beetroot powder has shown promise. These ingredients contain naturally occurring nitrates that can be converted into nitrites during curing, preserving antimicrobial properties and color. Efforts to develop guidelines and ensure sensory acceptance are ongoing, empowering consumers to make informed choices and promoting a healthier food industry.

## Figures and Tables

**Table 2 foods-12-02962-t002:** Approaches, tools, and implications in reducing or replacing salts in meat products.

Product	Approach	Tool	Implications	Reference
Harbin dry Sausage	Salt mixtures	SS1: 70% NaCl and 30% KCl; SS2: 70% NaCl, 20% KCl, 4% Lys, 1% alanine 0.5% citric acid, 1% Calactat, and 3.5% maltodextrin	Higher levels of methyl ketones, methyl and ethyl esters, methyl-branched aldehydes, and alcohols (70% NaCl, 20% KCl); technological properties unchanged.	[91]
Ham	Salt substitutes: reducing salt from 2.50% to 1.25%; Same + 0.2 Lys; same + 0.4 Lys; same + 0.6% Lys; same + 0.8% Lys		Increased cooking and centrifugation losses. Cooking loss tended to be reduced with increasing levels of Lys. Decrease in hardness, springiness, and chewiness. Improved the WHC and textural properties; improved the sensory scores for mouth-feel, appearance, taste, and global acceptance. Further impaired these attributes.	[86]
Dry-cured loin	Salt substitute: 0.2–0.4% of L-Lys; 0.05–0.4% of L-His		Delayed lipid oxidation, produced a reduction in sodium content; promoted physicochemical properties.	[82]
Bologna-type sausages	Salt substitutes: replacement of 60% NaCl by KCl; same + Arg and His alone or in combination		Formation of biogenic amines. Did not affect the color parameters. Arg and His were effective in reducing the defects caused by the addition of KCl. Emulsion stability, texture profile, and sensory quality were impaired by the salt substitution.	[22]
Jerked beef	Replacement of (50%) NaCl by 50% KCl (F1), 50% CaCl2 (F2), and a blend containing 25% KCl and 25% CaCl_2_		CaCl_2_ blends resulted in final product with bitter taste and rancid aroma and higher SF e TBARS; reduced sodium.	[72]
Dry-cured ham	Processing technology: high pressure application	500 MPa at 3ºC	Decrease softness in muscles.	[77]
Cooked ham	Processing technology: high pressure application	NaCl content (0, 0.95, 1.33, and 1.90%), phosphate content (0 and 0.25%) and the use of high-pressure processing (100, 300, 600 MPa)	More efforts are needed to understand the feasible levels of NaCl, phosphates, and specific HP treatments to apply in commercial scenarios. HPP at 100 MPa after tumbling was beneficial; salt reduction up to 1.1% NaCl was possible by the salt replacement with KCl (0.2%) in combination with the HP treatment.	[81]
Dry-cured pork bellies	Natural salty taste through sodium reduction	BC 100% NaCl BK had 50% of NaCl and 50% of KCl; BG had 90% of NaCl and 10% of GP.	BG in dry-cured pork bellies did not affect weight loss, a_W_, or pH. The sensory evaluation revealed differences in appearance, taste, and flavor among treatments, but did not indicate any negative effects of BG in the product attributes.	[92]
Cecina	Salt substitutes	50% NaCl; 50% KCl; 45% NaCl, 25% KCL; 20% CaCl_2_; 10% MgCl_2_	Non-significant differences in hardness. CaCl_2_ MgCl_2_ increased luminosity. Increased lipid oxidation and decreased redness (CaCl_2_ and MgCl_2_).	[93]
Lacón	Salt substitutes	50% NaCl; 50% KCl; 45% NaCl, 25% KCL; 20% CaCl_2_; 10% MgCl_2_ 30% NaCl, 50% KCL; 15% CaCl_2_; 5% MgCl_2_	50% NaCl; 50% KCl; and 30% NaCl, 50% KCl; 15% CaCl_2_; 5% MgCl_2_—lowest oxidation levels; 45% NaCl, 25% KCL;20% CaCl_2_; 10% MgCl_2_—highest degree of oxidation	[94]
Chicken sausage	Winter mushrooms (*Flammulina velutipes*) powder (WMP) as salt substitute	(1) positive control: 3 g/kg sodium pyrophosphate; (2) negative control: no sodium pyrophosphate nor WMP; (3) WMP 0.5: 5 g/kg of WMP; (4) WMP 1.0: 10 g/kg of WMP	Increased meat batter pH, reduction in jelly proportion and melted fat exuded. Texture softened. Lipid oxidation inhibited by WMP. No negative effects on color and sensory properties.	[95]
Chicken nuggets	NaCl substituted with CaCl_2_	Control (1.5 g NaCl/100 g) + 3 treatments with CaCl_2_ substituting 25, 50, 75% of NaCl	Similar sensory acceptance (texture, flavor and overall quality) between the control and 75% sodium reduction; decrease in salty taste in the just-about-right scale test.	[96]
Chicken nuggets	Salt replacement Added chickpea hull flour	1. Low salt (replacement 40% common salt) Low salt with chickpea hull flour	Affects appearance and flavor (1). Increased dietary fiber while decreased total cholesterol content (2).	[97]
Salt-baked-chicken	NaCl substituted with low sodium mixture (LS)	Control: 100% NaCl 2. LS: 52% NaCl + 30% KCl + 8% calcium lactate + 10% Yeast extract	Positive effect on taste and volatile flavor.	[98]

WHC—water-holding capacity; Arg—arginine; His—histidine; Lys—lysine; HP—high pressure; BC—belly control; BK—belly with NaCl and KCl; BG—belly with NaCl and GP; GP—glasswort powder; KCl—potassium chloride. LS—NaCl + KCl + Calcium lactate + Yeast extract.

**Table 3 foods-12-02962-t003:** Results and sensory implications of meat products processed with nitrite substitutes.

Meat Product	Nitrite Substitute	Result	Sensory Implication	Reference
Mortadella	Powdered parsley	Less growth of L. monocytogenes during storage and lower residual nitrite	Consumer acceptance for products produced with higher concentrations of parsley extract was given	[121]
Sausages	Pest powder and celery powder with high pressure (HP)	Extends shelf life and improves sensory features	Sausages with celery powder had better sensory quality; low sensory acceptance of sausages with pest powder	[122]
“Sucuk” a Turkish fermented beef sausage	Beetroot powder	Increasing the a* (red index) outcome in a desired red color during storage	Samples with beetroot powder were comparable to those of control with nitrite	[100]
Sausages	Celery powder	Protection from quality deterioration during storage and ensures microbiological safety	Sausages treated with paprika powder and blueberry powder were the best evaluated	[123]
Fermented Sausages	Vegetable powders (celery, beet, and spinach)	Vegetable origin, in powder or in juice, offers high potential as natural substitutes for nitrates and nitrites in processed meats	Sensory evaluation of samples with celery powder comparable with control (potassium nitrate)	[116]
Sausages	Jasmine tea extract combined with high pressure (HP)	Reduced the *C. perfringens* count during storage	No results	[124]
Fermented cooked sausages	Chitosan and radish powder	Improves the microbiological stability	Overall acceptability was influenced	[125]
Frankfurter-type sausages	ε-polylysine (ε-PL) or ε-polylysine nanoparticle (ε-PLN) combined with plants extracts	Improves shelf life	Sausages formulated with ε-PLN had higher sensory properties	[30]
Pork bologna	Beet powder and lentil flour	Using both components together had potential application as a sodium nitrite substitute	To increase consumer acceptability, technical improvement is required to reduce the red purge that is released	[118]

## Data Availability

The data used to support the findings of this study can be made available by the corresponding author upon request.

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
