# Peer review of "Novel Approaches to Improve Meat Products’ Healthy Characteristics: A Review on Lipids, Salts, and Nitrites"

_foods, 2023, doi:10.3390/foods12152962_

Round 1

Reviewer 1 Report

Although meat products are mentioned in the title of the paper, poultry products are not mentioned anywhere, which I consider a big shortcoming and it may mislead to the potential reader considering the information he can get reading this paper. Poultry meat and poultry meat products have a large share in the human diet. Due to the above, I believe that the work needs to be supplemented with the requested information before considering the publication of this paper.
Author Response

Reviewer 1

Although meat products are mentioned in the title of the paper, poultry products are not mentioned anywhere, which I consider a big shortcoming and it may mislead to the potential reader considering the information he can get reading this paper. Poultry meat and poultry meat products have a large share in the human diet. Due to the above, I believe that the work needs to be supplemented with the requested information before considering the publication of this paper.

Response: We would like to thank the reviewer for the suggestions on our manuscript.

As suggested by the reviewer we added information about poultry to enrich the manuscript. New references were added and marked in the manuscript with yellow colour.

Reviewer 2 Report

This review focusing on synthesizing novel strategies to render meat products safer and healthier is indeed interesting. The authors grouped and highlighted the different strategies applied on experiments performed on varied meat products claimed to be healthier and safer. 

While the authors mainly focused on the strategies to reduce the lipids, salt, and nitrites in meat products to make them healthier, an important link seems missing, particularly with regard to the carcinogenicity of red meat as discussed by the authors in the Introduction section. Indeed, many studies have directed to render meat healthier in terms of "less carcinogenic". I believe the authors should focus on this axis to bring novelty to their work that would be interesting for public health.

To answer this, the authors need to discuss about the mechanisms by which red meat promotes cancer before listing the possible strategies for reducing meat carcinogenicity (see the works of Denis Corpet and Fabrice Pierre from INRA France). These strategies would cover, but not limited to, adding natural antioxidant compounds in meat products to reduce the formation of 4-hydroxynonenal, adding ascorbic acid in nitrite-containing meat products to prevent the formation of nitrosamine, and adding calcium for heme-trapping to reduce meat carcinogenicity.

If the meat-cancer axis is well described in this manuscript, I believe this review would bring novelty for the readers. Thank you.

Author Response

Reviewer 2

This review focusing on synthesizing novel strategies to render meat products safer and healthier is indeed interesting. The authors grouped and highlighted the different strategies applied on experiments performed on varied meat products claimed to be healthier and safer.

While the authors mainly focused on the strategies to reduce the lipids, salt, and nitrites in meat products to make them healthier, an important link seems missing, particularly with regard to the carcinogenicity of red meat as discussed by the authors in the Introduction section. Indeed, many studies have directed to render meat healthier in terms of "less carcinogenic". I believe the authors should focus on this axis to bring novelty to their work that would be interesting for public health.

To answer this, the authors need to discuss about the mechanisms by which red meat promotes cancer before listing the possible strategies for reducing meat carcinogenicity (see the works of Denis Corpet and Fabrice Pierre from INRA France). These strategies would cover, but not limited to, adding natural antioxidant compounds in meat products to reduce the formation of 4-hydroxynonenal, adding ascorbic acid in nitrite-containing meat products to prevent the formation of nitrosamine, and adding calcium for heme-trapping to reduce meat carcinogenicity.

If the meat-cancer axis is well described in this manuscript, I believe this review would bring novelty for the readers. Thank you.

Response: We would like to thank the reviewer for the suggestions on our manuscript.

As suggested by the reviewer we added information about carcinogenicity to enrich the manuscript. New references were added and marked in the manuscript with yellow colour.

Reviewer 3 Report

The current paper reviews Novel approaches to improve meat products’ healthy characteristics. A review on lipids, salts, and nitrites

My observations can be found below:

Introduction.

Lines 26 to 60 requires more references to sustain the background problem of the present study.

Line 66 -71. Here the authors talk about enhancing meat lipid quality by enhancing the fats profile, however, references 5 to 16, are about sausages or pate. In my opinion this is not very accurate because is wheatear meat or meat products?! If we talk about meat quality, literature data revealed that plants (https://doi.org/10.3390/foods11081105), fats, oils (https://doi.org/10.1016/B978-0-08-100593-4.00023-0) and so on, are used as dietary ingredients to enhance meat quality, or while using ingredients directly into a food product like sausages or pate is a different strategy. Please rephrase to avoid confusions.

Line 84. Please provide values for nitrites limits.

Line 154. Table 1. The data presented in this table should be discussed by the authors a little bit. For example, in references 39, 41, and 8, lipid oxidation increased as a response to this fat manipulation, which resulted in a products more perishable being an unwanted effect.

Line 233. The study of … give the name (69).

Table 2, first line .. salt mixtures instead of mistures

Line 345 -353. What about the European regulations? Since these compounds are still allowed to be used as food conservers there is a problem with the applicable legislation. The authors consider that some changes in the current legislation are required to change their usage? As in the case of antibiotics there is a need to ban the use of nitrites and nitrates for example?

Also it is fesable to use L-lysine, L-histidine, and L-arginine to reduce salt in meat products? Are this AA accessible and cheap to be used as alternatives?

In my opinion the authors should discuss in more detail the legislation the consumers’ acceptance (https://doi.org/10.1016/j.cofs.2020.12.004) of these changes in the meat products and also the accessibility of the alternatives studied.

Author Response

Reviewer 3

The current paper reviews Novel approaches to improve meat products’ healthy characteristics. A review on lipids, salts, and nitrites

My observations can be found below:

Introduction.

Lines 26 to 60 requires more references to sustain the background problem of the present study.

Response: We would like to thank the reviewer for the suggestions on our manuscript.

As suggested by the reviewer we added information about health issues related with meat products consumption to enrich the manuscript. New references were added and marked in the manuscript with yellow colour.

Line 66 -71. Here the authors talk about enhancing meat lipid quality by enhancing the fats profile, however, references 5 to 16, are about sausages or pate. In my opinion this is not very accurate because is wheatear meat or meat products?! If we talk about meat quality, literature data revealed that plants (https://doi.org/10.3390/foods11081105), fats, oils (https://doi.org/10.1016/B978-0-08-100593-4.00023-0) and so on, are used as dietary ingredients to enhance meat quality, or while using ingredients directly into a food product like sausages or pate is a different strategy. Please rephrase to avoid confusions.

Response: We would like to thank the reviewer for the suggestions on our manuscript.

“products” was added to avoid confusions and make it accurate. We’re writing about “meat products” and not “meat”.

Line 84. Please provide values for nitrites limits.

Response: We would like to thank the reviewer for the suggestions on our manuscript.

Nitrites limits were provided in the manuscript.

Line 154. Table 1. The data presented in this table should be discussed by the authors a little bit. For example, in references 39, 41, and 8, lipid oxidation increased as a response to this fat manipulation, which resulted in a products more perishable being an unwanted effect.

Response: We would like to thank the reviewer for the suggestions on our manuscript.

A discussion was added in the manuscript as suggested by the reviewer.

Line 233. The study of … give the name (69).

Response: We would like to thank the reviewer for the suggestions on our manuscript.

The name of the authors was added as suggested by the reviewer.

Table 2, first line .. salt mixtures instead of mistures

Response: We would like to thank the reviewer for the suggestions on our manuscript.

“Mistures” was corrected to “Mixtures” as suggested by the reviewer.

Line 345 -353. What about the European regulations? Since these compounds are still allowed to be used as food conservers there is a problem with the applicable legislation. The authors consider that some changes in the current legislation are required to change their usage? As in the case of antibiotics there is a need to ban the use of nitrites and nitrates for example?

Response: We would like to thank the reviewer for the suggestions on our manuscript.

Information was added in the manuscript about European Regulations.

Also it is fesable to use L-lysine, L-histidine, and L-arginine to reduce salt in meat products? Are this AA accessible and cheap to be used as alternatives?

Response: We would like to thank the reviewer for the suggestions on our manuscript.

As far as it was possible to search for references it wasn’t possible to find one mentioning the real cost of using amino acids as salt replacers. Anyway, a text referring that the costs from the use of more healthy replacements can be a barrier to their use in the industry was added in the manuscript.

In my opinion the authors should discuss in more detail the legislation the consumers’ acceptance (https://doi.org/10.1016/j.cofs.2020.12.004) of these changes in the meat products and also the accessibility of the alternatives studied.

Response: We would like to thank the reviewer for the suggestions on our manuscript.

New information, marked in the manuscript, was added as suggested by the reviewer.

Round 2

Reviewer 1 Report

Some typos are found in text:

- line 34: ex-tending (correct - extending)

- line 310: pro-duction (correct - production)

Reviewer 3 Report

The Authors corrected, modified, and improved the paper quality. I have no further observations.